# Effect of Cold Brew Coffee Storage in Industrial Production on the Physical-Chemical Characteristics of Final Product

**DOI:** 10.3390/foods12203840

**Published:** 2023-10-20

**Authors:** Damian Maksimowski, Maciej Oziembłowski, Joanna Kolniak-Ostek, Marcelina Stach, Muhamad Alfiyan Zubaidi, Agnieszka Nawirska-Olszańska

**Affiliations:** 1Department of Functional Food Products Development, Wroclaw University of Environmental and Life Sciences, 50-375 Wrocław, Poland; damian.maksimowski@upwr.edu.pl (D.M.); maciej.oziemblowski@upwr.edu.pl (M.O.); 2Department of Fruit, Vegetable and Plant Nutraceutical Technology, Faculty of Biotechnology and Food Science, Wrocław University of Environmental and Life Sciences, 51-630 Wrocław, Poland; joanna.kolniak-ostek@upwr.edu.pl (J.K.-O.); marcelina.stach@upwr.edu.pl (M.S.); muhamad.zubaidi@upwr.edu.pl (M.A.Z.)

**Keywords:** coffee, ready-to-drink, post-processing changes, bioactive activity

## Abstract

The purpose of this article is to present information about changes in physical properties (pH, TA, and color) and chemical components with bioactive activity in cold brew coffee beverages, during storage before and after HTST processing. Coffee samples were tested using industrial technology (12,000 bottles per batch). The antioxidant activity of the samples was analyzed using ABTS and FRAP methods, the concentration of polyphenols was determined using the UPLC-MS chromatography coupled with mass spectrometry method, and microbiological tests were performed according to PN-ISO/PN-EN ISO standards. The pH value decreased during coffee storage, and the color changed significantly in brightness. Polyphenol concentrations were calculated in the range of 1800 to almost 3400 mg/L, and the antioxidant capacity for ABTS and FRAP reached the ranges of the results successively: 123–195 µMol/100 mL and 158–212 µMol/100 mL. It was observed that HTST pasteurization has a beneficial effect on preserving the beverage in microbiological terms. Also, a positive effect of the process on the release of chemical components responsible for bioactive properties from the beverage was observed, followed by a reduction in antioxidant activity during the first 90 days of storage and between 180 and 270 days during storage.

## 1. Introduction

International trade is widely recognized as a key driver of global economics. Due to this reason, coffee is an important commodity in the food industry and the second largest commodity worldwide [1]. Furthermore, widespread access to new information has also made consumers an important factor for researchers in new product development and analysts predicting upcoming trends in this sector. Within these trends, cold brew coffee significantly grows in the ready-to-drink category and has become one of the most popular drinks in specialty coffee. This market was valued at USD 321 million in 2017, it is expected to reach USD 1.37 billion by 2023, and it seems that cold brew will be one in six coffees bought in 2023 [2].

Regardless of the technique of brewing (i.e., full immersion, cold tower drip), cold brew coffee gained considerable consumer acceptance due to different sensory profiles and the smoother, less acidic, and less bitter profiles than the hot brewed counterparts [3]. From the consumers’ point of view, coffee extracted using the aforementioned method is sweeter than that brewed using the hot drip method, as well as the French Press [4]. Parameters such as pH or titratable acidity can be related to the characteristics of the beverage and are therefore usually checked during production, and color measurement is an additional control parameter. In the context of pH, cold brew producers must understand that parameters must be strictly controlled to prevent the risk of Clostridium botulinum, a deadly pathogen in hermetically sealed containers [5]. On the other hand, increased interest in health-promoting properties may establish a new trend in the coffee beverage sector to determine the bioactive properties of the final products using, i.e., basic spectrophotometric analyses, which may also be convenient for determining caffeine or controlling the extraction process during production. In the past decade, cold brew coffee has been usually consumed immediately after preparation or after short storage in refrigerated conditions. In order to maximize information on cold brew coffee and gather information available now on shelf-life studies and different sterilization methods (pasteurization, back-pressure sterilization, high-pressure short-term sterilization, membrane filtration, and high-pressure processing) [6,7], the physicochemical changes in the beverage during spoilage were investigated based on the final product produced using HTST pasteurization, from the largest artisanal cold brew coffee producer in Poland (which appears to be the largest in Central Europe). Although the commonly used pasteurization conditions ensure microbiological safety, they also lead to complete or partial degradation of the biologically active ingredients. In this research, it was possible to compare selected parameters of an unpasteurized product, which is not available on the market due to microbiological risks. Caffeine was overlooked in the study because it was proven to be very stable during storage [8].

## 2. Materials and Methods

### 2.1. Reagent and Standard

Acetonitrile, formic acid, methanol, ABTS (2,2′-azinobis-(3-ethylbenzothiazoline-6-sulfonic acid), Trolox (6-hydroxy-2,5,7,8-tetramethylchroman-2-carboxylic acid), TPTZ (2,4,6-tri(2-pyridyl)-s-triazine were purchased from Sigma-Aldrich (Steinheim, Germany). All standard compounds for the characterization of phenolic compounds using LC–MS were purchased from Extrasynthese (Lyon, France). Sodium hydroxide and ascorbic acid were from Merck (Darmstadt, Germany).

### 2.2. Cold Brew Coffee Extract

The materials for the study consisted of cold brew coffee drinks purchased directly from the manufacturer ETNO Café in Wroclaw (Lower Silesia in Poland). In accordance with the company’s HACCP rules, all beverages were bottled in a 220 mL amber glass and made exclusively from Arabica coffee from Brazil, which was roasted in a medium-roasted profile. Coarsely ground beans with a diameter of 1.5–2.0 mm were mixed with reverse osmosis water in a corrosion-resistant stainless-steel tank equipped with a 3.0 kW overhead mixer. The extraction process by soaking was carried out until 1.3% tds was obtained under 12 h, followed by filtration through a 200 µm pore-width cartridge filter and HTST pasteurization. During production, the professional REFRACTOMETER ATAGO PAL-COFFEE BX/TDS tool was used to determine whether coffee extraction is processed exactly under the optimal equilibrium according to Lockhart’s diagram (Figure 1). The capacity of the technology reaches 3000 L of cold brew coffee per batch.

Five research variants were prepared and are shown in Table 1.

Variant one (CB1) was taken from a sampling valve located downstream of the filtration stage and prior to the pasteurization process, and was stored until testing at −18 °C. The second variant of the beverage sample (CB2) was pasteurized and bottled according to the established quality standards on the processing line and stored until testing at −18 °C. Variant three (CB3) was a beverage pasteurized and bottled according to the assumed quality standards on the processing line, which was first subjected to storage in an aging context for a period of 90 days at room temperature of 25 °C (±1 °C). After this, it was frozen to −18 °C until the time of the analyses performed. The fourth variant (CB4) was an analogy of the previous one, except that it was stored for 180 days at room temperature. The fifth variant (CB5) was pasteurized and stored at room temperature for a period of 270 days. All samples were from a single production batch.

### 2.3. Analysis of pH and Titratable Acidity

The pH of each brewed coffee sample was measured with a Mettler Toledo FiveEasyTM F20 (Mettler-Toledo, Warszawa, Poland) benchtop pH/mV meter.

The method of determination of titratable acidity consists of neutralization of the organic acids present in the tested coffees in a properly prepared solution of the tested sample with 0.1 M sodium hydroxide solution until the titration point is reached [9]. Approximately 10 mL of the test material was measured and transferred into 150 cm^3^ and 200 cm^3^ volumetric flasks and made up to 2/3 of the flask’s capacity with distilled water. The samples were boiled in a water bath for 30 min at 100 °C. After cooling the volumetric flasks to room temperature, they were again made up to the mark with distilled water. The resulting solutions were then mixed and filtered through material strainers, and 50 cm^3^ of each filtrate was taken. The obtained filtrates were titrated with 0.1 M sodium hydroxide solution using a titrator. The result was calculated by converting to malic acid content.

### 2.4. Analysis of Antioxidant Activity

Samples for ABTS antioxidant activity analysis were prepared as follows: 5 mL of coffee was centrifuged for 5 min. After the centrifugation, 0.03 mL of supernatant was taken for the spectrophotometric vial, and 3 mL of ABTS solution was added. The ABTS+ radical scavenging activity of the sample was measured using a Shimadzu UV-2401 PC spectrophotometer (Kyoto, Japan) followed by 6 min of incubation with the test extracts. The results were expressed as μmol Trolox per 100 mL. A standard curve was prepared using different concentrations of Trolox. Determinations of all traits were performed in triplicate using a Shimadzu UV-2401 PC spectrophotometer (Kyoto, Japan). Compounds with antioxidant properties in these extracts reduced the concentration of the cation radical ABTS+, which was measured as the decrease in absorbance of the solution at 734 nm. The results were calculated by the difference in absorbance value versus blank [10].

The reducing potential of the sample was determined using the FRAP assay [11] as a measure of antioxidant power. An antioxidant reduces the ferric ion (Fe^3+^) to the ferrous ion (Fe^2+^); the latter forms a blue complex (Fe^2+^/TPTZ), which increases the absorbance at 593 nm. A standard curve was prepared using different concentrations of Trolox. The results were corrected for dilution and expressed as µmol Trolox per 100 mL of coffee. Determination of all characteristics was carried out in triplicate using a Shimadzu UV-2401 PC spectrophotometer (Kyoto, Japan).

### 2.5. LC-MS Analysis of Polyphenols

Analysis was started from the centrifugation of all samples at 19,000× *g* for 10 min, and the supernatant was filtered through a hydrophilic PTFE 0.20 µm membrane (MillexSimplicity Filter, Merck, Rahway, NJ, USA) and used for analysis.

Identification of polyphenols of coffee extracts was carried out using an ACQUITY Ultra Performance LC system equipped with a PDA detector (Waters Corporation, Milford, MA, USA) with a mass detector G2 Q-TOF micro mass spectrometer (Waters, Manchester, UK) equipped with an ESI source operating in negative mode. A UPLC BEH C18 column (1.7 µm, 2.1 mm × 100 mm, Waters Corporation, Milford, MA, USA) was used. Samples (10 µL) were eluted according to the linear gradient described previously by Oziembłowski et al. [12]. Each compound was optimized to its estimated molecular mass in the negative mode, before and after fragmentation, and monitored at 320 nm. All experiments were conducted in triplicate. The results were expressed as milligrams per 1 L of extract.

### 2.6. Microbial Analysis

Due to the nature of the implementation-scientific work, microbiological studies were performed for variants 2–5. Variants constituted a commercial product, as they were thermally stabilized (using HTST technology by electric pasteurizer Gebhardt Anlagen Technik GmbH & Co. KG, Herbolzheim, Germany, model EHA45).

Analyses were performed using the plate method (surface culture) on behalf of the Etno Cafe Company in an accredited laboratory working according to PN-ISO Polish Standards [13]. Therefore, test results lower or higher than the measurement ranges of the methods are presented as “<value of the lower limit of the measurement range” or “>value of the upper limit of the measurement range”, respectively. These values provide information about the results of testing three samples for each variant. The results can be used to assess compliance in the regulatory area.

The second selective substrate for detecting the presence of *Salomonella* spp. was RVS broth and Brilliance Salomonella/Agar. The temperature and incubation time used for coagulase-positive staphylococci was 37 ± 1 °C for 48 ± 4 h. The applied temperature for incubation of coli form bacteria was 37 ± 1 °C.

### 2.7. Color Analysis

The color of the beverages was determined with an on-camera instrumental method on the ColorQuest (HunterLab, Reston, VA, USA) apparatus, on the L*a*b* scale, D65 illuminant, and observer 100, transmitted light. The color of the coffee was additionally measured after 3, 6, and 9 months of storage.

### 2.8. Statistics

The obtained data were subjected to statistical analysis performed using Statistica v.10.0 (StatSoft Polska, Kraków, Poland). They were recorded as means ± standard deviation (SD) and analyzed by the Microsoft Excel 2007 software (Microsoft Corp., Redmond, WA, USA). Analysis of variance was performed with ANOVA procedures. Significant differences (*p* ≤ 0.05) between mean values were determined by Duncan’s multiple-range test.

## 3. Results

### 3.1. Titratable Acidity and pH

During roasting, green coffee beans undergo several changes (e.g., pyrolysis, Maillard reaction, Strecker degradation, and caramelization) leading not only to the development of characteristic properties of the coffee beverage such as flavor, aroma, and color changes but new compounds with strong antioxidant properties are also formed, which may additionally alter perceived acidity [14]. The pH of the tested coffees ranged from 4.25 to 4.67. In a similar study [15], pH values ranged from 5.40 to 5.63, but it may be due to the different technological processing of the test samples and the different varieties of tested coffee. The highest values of pH were observed in both fresh coffees CB1 (without pasteurization) and CB2 (directly after pasteurization). Interestingly, the pH parameters of CB2 were not statistically different from the results for sample CB5. On the other hand, measured acidity was highest in coffee after 3 months of storage and lowest in fresh coffee after 9 months of storage. Results obtained for CB5 seem to be an interesting report and require further research because a correct relationship between pH and TA parameters was observed only between samples CB1 through CB4—as TA increased as pH decreased (Table 2). The last sample CB5 indicates a deviation from the mentioned rule; TA was not significantly different from sample CB1. This phenomenon can be attributed to the beginning of the spoilage process. The positive results of the pasteurization process were also confirmed by Bellumori et al. [6]. Cold brew coffee samples after 120 days of storage have been able to guarantee the safety of the beverage for up to four months.

### 3.2. Antioxidant Capacity

Today, coffee is also a very popular drink consumed by many people due to its rich source of non-enzymatic bioactive ingredients, as well as due to its health-promoting properties, with noticeable antioxidant properties related to content esters of quinic acid and caffeic acid and their derivatives [16]. Researchers often used ABTS assay to evaluate this due to its applicability to beverages containing also high pigmented antioxidant compounds [17], especially for hot coffees where high temperature could faster solubilities of compounds with higher molecular weight.

Muzykiewicz-Szymańska et al. [10] proved very high antioxidant activity ranging up to 99.43% RSA (roasted coffee). According to Kang et al. [18], cold brewing methods significantly increased the ABTS, whereas the data of analyzed CQA concentrations along with isomers by Rao et al. [19] indicate that the water extraction temperature does not significantly differentiate the total content of compounds between cold brew and hot brew. On the other hand, Stanek et al. [20] in six different arabica beverages compared the total of phenolic compounds, coupling the results with the ABTS activity, and concluded that high temperature appears to be a key determinant of the extraction efficiency of antioxidant compounds, which we agree with. It is noteworthy that the low temperature and prolonged time may compensate for the bioactive properties of the beverage obtained by cold brewing, especially on coffee from Brazil, as the authors showed no significant differences between the total phenolic compounds and the ABTS activity compared to hot brews.

Pasteurized coffee had a slightly higher antioxidant activity of about 10% than fresh coffee (Table 3). These values are in agreement with the results of Pokorna et al. [21], whose antioxidant activity measurements also showed slightly higher values for the medium roast profile compared to light roast in *C. arabica* extracts. Along with other results [22] on the antioxidant capacity of coffee infusions, it was observed that antioxidant activity can increase depending on the brewing method but only increases from the light to medium roast profile of green coffee. A high effect on the decrease in antioxidant activity and a disproportion can be seen with a dark roasting profile at 230 °C. Results from Schwarzmann et al. [23] have shown that hot brew methods exhibit more effective radical scavenging activity than cold brew methods, but the differences in TAC values between samples can be relatively small. Hence, it was observed that the possibility of maintaining or even slightly increasing the antioxidant properties of cold brew coffee in the study resulted from advanced pasteurization. Compounds with antioxidant activity may have been additionally released from the structures of endiol or Maillard reaction products that react with ABTS•+/FRAP•+. It seems that further studies using liquid chromatography as a complementary method are needed.

Generally, the activity decreased during storage. It can be assumed that a significant reduction in antioxidant activity occurs in the first three months, and for the next three, it remains at a similar equal level (6 months of storage). In the ninth month of storage, another significant reduction in activity is observed. Antiradical activity against ABTS indicated for fresh cold brew coffee was 163 µMol trolox/100 mL, while the reducing capacity was lower and significantly different compared to fresh coffee pasteurization. For CB2, it was 195 µMol trolox/100 mL (*p* < 0.05). ABTS activity in cold brewed coffees three and six months after production ranged from 155 to 151 µMol trolox/100 mL, but these values were averaged 10 µMol lower than in unprocessed samples (*p* < 0.05). Interestingly, TAC of coffee has been studied extensively in Peaberry coffee extracts, which are esteemed for their characteristic properties (mutation of coffee beans where coffee berries contain one rounded seed instead of the usual two) [23]. Obtained results were ten times lower, but the ratio between coffee to water was almost two times higher comparable to the mentioned review. Furthermore, the literature review also indicates that the hot brewing technique does not necessarily have a positive effect on obtaining infusions with higher antioxidant potential than the cold method [24].

### 3.3. Characterization of Phenolic Compounds by LC–MS

Phenolic compounds are a heterogeneous group of secondary metabolites in plants. In coffee, the most dominant is chlorogenic acid, and it is isomers containing 69 structures in green coffee beans, but the major CGA isomers found in coffee include 3-caffeoylquinic acid (3-CQA), 4-caffeoylquinic acid (4-CQA), and 5-caffeoylquinic acid (5-CQA) [25]. Figure 2 and Table 4 list the 15 compounds identified in experiments using UPLC-MS/MS (with PDA and Q/TOF detectors) along with their retention times (Rt), UV-vis mass profiles of authentic standards, and concentrations per sample in terms of mg/L. All compounds analyzed were reported previously in coffee beans [26,27].

The importance of CGA in cold-brewed coffee, especially in products produced in large quantities in the context of the aging process, is a challenge, as no such detailed results are available in the literature. Analyzing samples and implementing them in the next brewing process may establish a more beneficial effect of extracts in the future. In non-stored coffees, 3-O-caffeoylquinic acid was the most determined at 1223.18 mg/L in fresh extract and 1235.39 mg/L after pasteurization (Table 5). At the same time, this acid was found to be the least stable during the storage process, with a 98.5% decrease in its content at the end of the study period compared to fresh coffee. It may be considered to be the main derivative of phenolic compounds responsible for antioxidant capacity in coffee, so TAC significantly decreased after long storage. Another compound that was determined in significant amounts was 5-O-caffeoylquinic acid. In the fresh extract and after pasteurization, 5-O-feruloylquinic acid and 4-O-p-coumaroylquinic acid were determined, which were not determined in the coffees after storage. In contrast, 5-O-p-coumaroylquinic acid and cis-3-O-feruloylquinic acid, which were not labeled in the fresh coffees, were labeled in small amounts in the coffees after storage. It can therefore be concluded that there was a conversion of 5-O-feruloylquinic acid to small amounts of cis-3-O-feruloylquinic acid and 4-O-p-coumaroylquinic acid to 5-O-p-coumaroylquinic acid during storage. While the content of 5-O-p-coumaroylquinic acid decreased during storage, the cis-3-O-feruloylquinic acid increased.

As shown, thermal preservation (HTST—high temperature short time) increased CGA value significantly, reaching 3392.61 mg/L, and it is not in agreement with Bellumori et al. [6]. It may be due to LTLT processing, which is less sophisticated than HTST, but after 120 days, the concentration of CGA compounds was reported only for the pasteurized and HPP-treated samples. Furthermore, our data showed close to 380 mg/L higher values in non-treated cold brew coffee, without using the dedicated extraction commercial brewing system Cold Pro™ (Brewista, Brea, CA, USA).

On the other hand, the results of the present study demonstrate that the proportion and level of phenolic acids varied greatly among cold brew samples. The large disproportion between samples immediately after pasteurization and matured nine months samples can reach close to 1600 mg/L, which may indicate high degradation by oxidation. The key mechanism of its degradation appears to be governed by the polyphenol oxidase in the coffee because PPO is a copper-containing enzyme responsible for the hydroxylation of and oxidation of phenols, as well as established that low-quality coffee has low PPO activity [28]. Generally, the analysis showed that the total amount of chlorogenic acid and isomers in fresh cold brew coffee was 3279.50 mg/L and decreased to 1804 mg/L after nine months, whereas results obtained by Rao and Fuller [29] indicate that total CQA reached 2503 mg/L, and 5-CQA was found to have the highest concentration 1261 mg/L in Brazilian samples. In turn, in similar research [30], the most abundant CGA isomer was 5-CQA, which accounted for 69% to 74% of total CGA, but it was in hot brew coffee extracts. We suppose that the development of the study could also agree with an interesting comparison of chlorogenic acid concentrations in four coffee brewing techniques (three hot and one cold), where the cold brewing method gave the highest (0.162 mg/mL), while the espresso infusion gave the lowest value (0.143 mg/mL) [24].

### 3.4. Microbial Analysis

Many studies confirm that the HTST process inactivates bacterial flora [6,31,32,33]. Table 6 shows the microbiological results. Treatment of the beverage with HTST technology resulted in the protection of the samples from microorganisms. The study shows that the use of HTST is effective in protecting beverages from microbial contamination while maintaining bioactive properties. However, when developing a new product, the effect of pasteurization on other physicochemical indicators of quality should be taken into account, since the process is not inert to aroma and taste.

### 3.5. Color Analysis

As shown in Table 7, coffee becomes darker after the HTST process and over time. Fresh coffee had the lightest color, while coffee stored for 9 months had the darkest color. The differences are significant as shown by the statistical analysis. The proportion of red color was highest in coffee stored for 9 months and in coffee after pasteurization. The analyses showed that the proportion of green color was also significant and the lowest was in the coffee after storage for 3 and 6 months. The proportion of yellow color followed the same pattern as red color, forming the same homogeneous groups, with the proportion of blue color in all coffees being lower than that of green color.

In the context of processing, the data are in line with Zang [34], who presented a significant effect of HPP treatment on the L-value during cold brew extraction from whole coffee beans. An almost twofold decrease in brightness was observed. In some cases, without special ingredients to correct this factor food processing is not inert to color, and differences in coffee beverage seems to be difficult to detect visually without specific additives or milk-type ingredients. On the other hand, coffee containing phenolic groups are known to have a general tendency towards changing color. If aging is taking place, it has an effect on spontaneous oxidation processes, commonly considered an undesirable factor for color, as can be seen from the decrease in antioxidant capacity between samples CB2–CB5.

Figure 3 shows the shade of color of the coffees tested. It can clearly be seen that the lightest is fresh coffee and the darkest is coffee stored for 9 months. In contrast, no difference can be seen between coffees stored for 3 and 6 months. This coincides with the data in Table 7. After storage for 3 and 6 months, there was little difference between the brightness and the proportion of red and yellow color. Both variants belonged to one homogeneous group. Coffee stored for 9 months had the highest proportion of yellow and red bars with the lowest brightness.

## 4. Conclusions

The effect of storage time on the physicochemical properties of cold-brewed coffee has been proven. Pasteurization affects the concentration of CGA in cold coffee extract and can maintain or even increase the antioxidant capacity immediately after production. Between 0 days and 270 days of storage, phenolic compounds are broken down, and differences reach up to 1530 mg/L. The analysis showed that during 90-day storage, the content of phenolic acids reached above 2700 mg/L, which corresponds to only 20% weaker antioxidant properties compared to the sample obtained immediately after pasteurization. In addition, the HTST technique effectively controlled microorganisms throughout the storage period. pH, TA and color parameters measured on the last day of storage may indicate the beginnings of undesirable changes, including spoilage of the product in the package. Based on the results obtained, it can be concluded that cold brew coffee within three months of production is the best option as part of a healthy diet. Limitations of tests regarding sensory properties were noticed in the storage context, so in this field, further research may be recommended.

## Figures and Tables

**Figure 1 foods-12-03840-f001:**
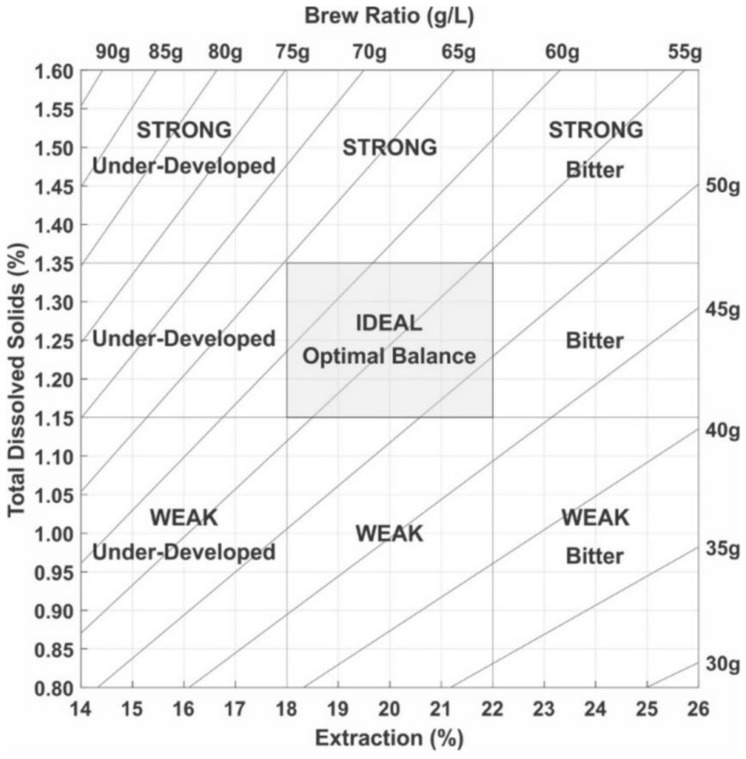
Coffee Lockhart’s diagram.

**Figure 2 foods-12-03840-f002:**
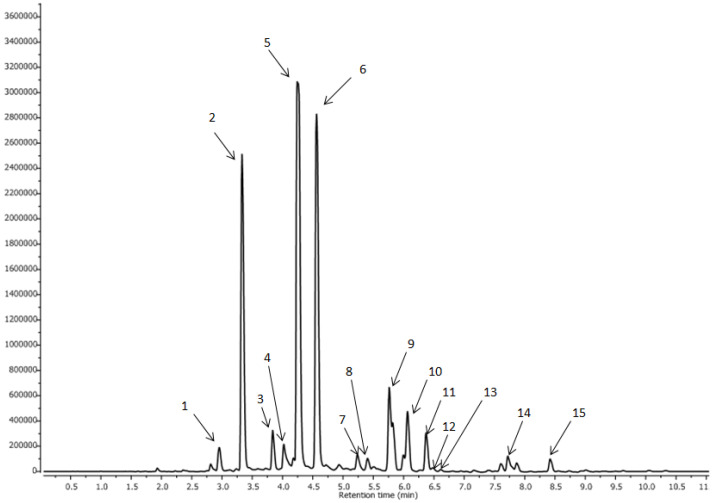
UPLC–MS chromatogram profile of coffee extracts at 320 nm. Peak number identities are displayed in Table 4.

**Figure 3 foods-12-03840-f003:**
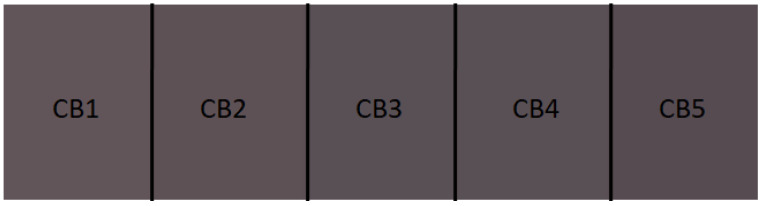
Hue of the tested cold brew coffees.

**Table 1 foods-12-03840-t001:** Research variants.

Variant	HTST Pasteurization	Storage Time at Room Temperature	Frozen Time
CB1	not pasteurized	not stored	270 days
CB2	pasteurized	not stored	270 days
CB3	pasteurized	stored 90 days	180 days
CB4	pasteurized	stored 180 days	90 days
CB5	pasteurized	stored 270 days	not frozen

**Table 2 foods-12-03840-t002:** Titratable acidity and pH of cold brew samples.

Variant	pH	Titratable Acidity (TA) *
CB1	4.67 ± 0.01 ^a^	0.10 ± 0.00 ^d^
CB2	4.33 ± 0.00 ^b^	0.12 ± 0.0 ^c^
CB3	4.25 ± 0.00 ^c^	0.16 ± 0.01 ^a^
CB4	4.28 ± 0.01 ^c^	0.14 ± 0.01 ^b^
CB5	4.32 ± 0.00 ^b^	0.11 ± 0.00 ^d^

Designation of test samples: CB1—untreated HTST and frozen 270 days; CB2—HTST processed and frozen 270 days; CB3—HTST processed and stored 90 days followed by frozen 180 days; CB4—HTST processed and stored 180 days followed by frozen 90 days; CB5—HTST processed and stored 270 days; mean values with different letters a–d were statistically different (*p* = 0.05); values expressed as mean ± standard deviation; *—mg/g malic acid.

**Table 3 foods-12-03840-t003:** Antioxidant activity in cold brew coffee samples.

Variant	ABTS [µMol/100 mL]	FRAP [µMol/100 mL]
CB1	163.33 ± 12.67 ^b^	197.89 ± 16.32 ^b^
CB2	195.00 ± 17.98 ^a^	212.04 ± 19.47 ^a^
CB3	151.67 ± 12.34 ^c^	176.72 ± 12.65 ^c^
CB4	154.33 ± 12.17 ^c^	178.38 ± 14.39 ^c^
CB5	123.67 ± 11.56 ^d^	158.16 ± 15.22 ^d^

Designations are shown in Table 1; mean values with different letters a–d, values with *p* = 0.05 were statistically different; values expressed as mean ± standard deviation.

**Table 4 foods-12-03840-t004:** Groups of CQA compounds identified by LC-PDA-ESI-MS/MS in coffee.

No.	Compounds	Rt (min)	[M-H]-	MS-MS
1	3,4-di-O-caffeoylquinic acid	2.95	515.143	353.096; 191.068;
2	3,5-diO-caffeoylquinic acid	3.30	515.145	353.096; 191.066
3	1-O-caffeoylquinic acid	3.83	353.095	191.096; 179.042
4	3,4,5-tri-O-caffeoylquinic acid	4.02	677.1816	353.0576, 191.0683
5	3-O-caffeoylquinic acid	4.25	353.096	191.069; 179.048; 135.059
6	5-O-caffeoylquinic acid	4.56	353.096	191.069
7	3-O-feruloylquinic acid	5.24	367.1126	191.0689
8	3-O-p-coumaroylquinic acid	5.40	337.0996	191.0685
9	cis-3,5-di-O-caffeoylquinic acid	5.76	515.121	353.096; 193.062;
10	5-O-feruloylquinic acid	6.06	367.112	193.064; 173.059
11	4-O-p-coumaroylquinic acid	6.37	337.087	161.038; 135.059
12	5-O-p-coumaroylquinic acid	6.47	337.0968	191.0697
13	cis-3-O-feruloylquinic acid	6.59	367.1129	191.0689
14	4,5-di-O-caffeoylquinic acid	7.71	515.12	353.094; 191.069
15	cis-4,5-di-O-caffeoylquinic acid	8.41	515.114	353.098; 191.069

**Table 5 foods-12-03840-t005:** Comparison of phenolic compounds detected in cold brew coffee [mg/L].

No.	CB1	CB2	CB3	CB4	CB5
1	55.09 ± 2.19	88.14 ± 4.36	153.72 ± 9.89	92.90 ± 3.45	75.50 ± 5.32
2	599.61 ± 23.76	692.44 ± 46.72	843.24 ± 67.33	663.03 ± 23.78	597.92 ± 21.89
3	70.18 ± 4.91	77.82 ± 4.66	130.12 ± 11.72	79.65 ± 5.61	86.10 ± 4.46
4	34.03 ± 1.77	40.59 ± 1.98	52.56 ± 2.55	39.80 ± 2.44	33.69 ± 1.21
5	1223.18 ± 99.45	1235.39 ± 11.34	28.28 ± 14.77	27.23 ± 1.46	18.55 ± 1.17
6	986.27 ± 51.28	1027.53 ± 10.02	1199.62 ± 9.54	987.94 ± 7.89	821.82 ± 62.44
7	13.82 ± 12.44	12.71 ± 11.12	19.99 ± 17.36	13.03 ± 11.14	10.57 ± 9.52
8	14.36 ± 1.23	14.22 ± 1.22	25.87 ± 1.33	15.75 ± 1.12	18.98 ± 1.61
9	151.86 ± 14.22	148.01 ± 14.63	216.76 ± 19.44	143.64 ± 11.38	103.50 ± 9.98
10	69.76 ± 5.54	14.45 ± 1.44	0.00	0.00	0.00
11	24.41 ± 1.99	2.33 ± 0.22	0.00	0.00	0.00
12	0.00	0.00	1.53 ± 0.12	0.63 ± 0.05	0.82 ± 0.06
13	0.00	0.00	0.78 ± 0.05	1.54 ± 0.44	1.68 ± 0.53
14	13.62 ± 1.33	16.22 ± 1.18	8.34 ± 0.72	15.42 ± 1.13	32.05 ± 2.22
15	23.29 ± 2.15	22.74 ± 2.12	26.13 ± 2.53	19.12 ± 1.19	3.59 ± 0.26
sum	3279.50	3392.61	2706.96	2099.67	1804.77

**Table 6 foods-12-03840-t006:** The microbial analysis of tested cold brew coffee.

No.	Tested Parameter	Unit	Research Methodology PN-ISO/PN-EN ISO	Variant	Result
1	Molds	CFU/mL	PN-ISO 21527-1:2009	CB2	<1.0 × 10^0^
				CB3	
				CB4	
				CB5	
2	Yeasts	CFU/mL	PN-ISO 21527-1:2009	CB2	<1.0 × 10^0^
				CB3	
				CB4	
				CB5	
3	*Salomonella spp.*	in 25 mL	PN-EN ISO 6579-1:2017-04 PN-EN ISO 6579-1:2017-04/A1:2020-09	CB2	Not detected
				CB3	
				CB4	
				CB5	
4	Number of coagulase-positive *Staphylococus aureus*	CFU/mL	PN-EN ISO 6888-2:2022-03	CB2	<1.0 × 10^0^
				CB3	
				CB4	
				CB5	
5	Presence of coli form bacteria	in 0.1 mL	PN-ISO 4831:2007	CB2	Not detected
				CB3	
				CB4	
				CB5	
6	Total microbial count	CFU/mL	PN-EN ISO 4833-1:2013-12 PN-EN ISO 4833-1:2013-12/Ap1:2016-11	CB2	<1.0 × 10^0^
				CB3	
				CB4	
				CB5	
7	Number of *Listeria monocytogenes*	CFU/mL	PN-EN ISP11290-2:2017-07	CB2	<1.0 × 10^0^
				CB3	
				CB4	
				CB5	

**Table 7 foods-12-03840-t007:** The brightness and color of cold brew coffee.

Variant	L	a	b
CB1	24.65 ± 1.23 ^a^	0.67 ± 0.03 ^c^	1.12 ± 0.15 ^c^
CB2	23.36 ± 1.45 ^b^	0.91 ± 0.04 ^b^	1.38 ± 0.23 ^b^
CB3	22.33 ± 1.12 ^d^	0.61 ± 0.01 ^d^	1.05 ± 0.17 ^d^
CB4	22.23 ± 1.17 ^d^	0.60 ± 0.01 ^d^	1.07 ± 0.11 ^d^
CB5	21.28 ± 1.06 ^e^	0.98 ± 0.09 ^a^	1.47 ± 0.13 ^a^

Designations are indicated in Table 1; mean values with different letters a–e were statistically different (*p* = 0.05); values expressed as mean ± standard deviation.

## Data Availability

Data is contained within the article.

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
