# Peer review of "Effect of Cold Brew Coffee Storage in Industrial Production on the Physical-Chemical Characteristics of Final Product"

_foods, 2023, doi:10.3390/foods12203840_

Round 1

Reviewer 1 Report

Comments and suggestions are provided in the attached file.

Author Response

All answers are in the attached file.

Reviewer 2 Report

Modifications, additions and revisions that must be made:

Abstract

The Abstract do not describe the complete research and the rest of paper. Why pH and color were tested at all, if they were not mentioned in the Abstract? Do these features have any relation to the quality of final product?

Furthermore, the last sentence of the Abstract seems not completely correct:

„Storage of products under 23 room temperature conditions for 9 months did not significantly affect the quality of the tested coffee.“

If the changes that occucred in product, that were observed during 9 months of storage and described in the paper, does not decrease the quality, authors should explain it somewhere in paper – what exactlly means „quality“(e.g. absence of pathogens?). It seems to me that after 9 months of storage samples had different color and different contents….

Line 29 – Statement „ coffee is the most important commodity in food industry“

Is it true?

Lines 188-192 – „The general statement only in the 188 context of the brewing process that cold-brewed coffee is "less acidic" compared to hot-189 brewed coffee is not correct from our point of view, as we see many factors related to 190 coffee roasting that affect this perception. The general perception may be that it is sweeter 191 in flavor, but a beverage prepared from ground coffee and water alone will always have 192 a slightly acidic pH independent from preparing methods.“

How „from our point of view“  can be proved in scientific paper? Some measurements?

Lines 220-221 – „ We think that coffee brewing using hot water could affect the different 220 solubilities of compounds with…“

How statement „we think“ can be proved in scientific paper? Some measurements?

Figure 3 – unclear!

Chapter conclusion – it is unclear why pH and color were investigated, since they were not mentioned in Conclusion. Do they reflects the quality of product in any way?

f

Author Response

All answers are in the attached file.

Reviewer 3 Report

The topic covered in coffee processing is interesting. I would like to highlight:

line338-339: in figure 3, no differences in color can be distinguished, as the authors state.  It is necessary to improve the quality of the figure so that one can appreciate what is stated in the text.

Author Response

All answers are in the attached file.

Reviewer 4 Report

Comments for the authors:

1. The authors are encouraged to improve the abstracts section by inserting more key-findings.

2. The quality of Figure 1 and Table 5 (seems to be added as jpg) has to be improved.  

3. Please, delete horizontal lines between raws in the tables 1-6. The letters a–d in the tables should be in superscript.

4. The manuscript should be revised for the grammatical errors and typos.

Author Response

All answers are in the attached file.

Reviewer 5 Report

Dear authors, 
The article entitled "The effect of cold brew coffee storage in industrial production 2 on the quality of final product" covers an interesting topic. However, this article has a fundamental flaw. My comments are below:

1. This article intended to study the effect of storage. This study, however, only show the data of Day 270. I cannot find (at least) the data of Day 0. It will be much better if the author also collect the data of (e.g.) Day 90 and Day 180. This article, therefore, seems only show the characteristic of the coffee at the end of the storage period, but not show the effect of storage at all. 

2. Abstract: The result mentions HPST, but it was not mentioned in the problem statement. 

3. Introduction: Some parameters analyzed in this research must be be introduced in the background of study (i.e., antioxidant, colour, etc). 

4. Introduction: Please emphasize the novelty of this research. There are some similar studies that have been published in various scientific databases. Therefore, please refer those studies as the background. 

5. Section 2.2: Please summarize CB1-CB5 in a Table to increase the readability 

6. Result and Discussion: Data on Day 270 must be compared with (at least) Day 0

7. Conclusion: Please add the recommendation for future studies based on the result obtained in this study and the limitation of this study. 

Some grammatical error and miss-typing are detected. Please check it carefully. 

Author Response

All answers are in the attached file.

Round 2

Reviewer 5 Report

The authors have improved the article based on the comments.